# Macro-TSH: An Uncommon Explanation for Persistent TSH Elevation That Thyroidologists Have to Keep in Mind

**DOI:** 10.3390/jpm13101471

**Published:** 2023-10-08

**Authors:** Isabella Chiardi, Mario Rotondi, Marco Cantù, Franco Keller, Pierpaolo Trimboli

**Affiliations:** 1Clinic for Endocrinology and Diabetology, Lugano Regional Hospital, Ente Ospedaliero Cantonale, 6500 Bellinzona, Switzerland; 2Faculty of Medicine and Surgery, Humanitas University, 20089 Milan, Italy; 3Department of Internal Medicine and Therapeutics, University of Pavia, 27100 Pavia, Italy; 4Unit of Endocrinology and Metabolism, Laboratory for Endocrine Disruptors, Istituti Clinici Scientifici Maugeri IRCCS, 27100 Pavia, Italy; 5Institute of Laboratory Medicine, Ente Ospedaliero Cantonale, 6500 Bellinzona, Switzerland; 6Faculty of Biomedical Sciences, Università della Svizzera Italiana (USI), 6900 Lugano, Switzerland

**Keywords:** TSH, thyrotropin, macro, PEG, interference

## Abstract

A macro-thyroid-stimulating hormone (macro-TSH) is an infrequent yet noteworthy phenomenon in the thyroid field. A 69-year-old patient presented with persistently elevated thyroid-stimulating hormone (TSH) levels ranging from 30 to 50 mIU/L, paradoxically accompanied by normal thyroid hormone levels and normal thyroid ultrasound, with no findings on pituitary magnetic resonance. Laboratory studies were conducted to investigate potential interferences impacting the accuracy of TSH measurements. After excluding other potential causes, polyethylene glycol (PEG) precipitation technique was used, which led us to the diagnosis of macro-TSH. This result was confirmed through chromatography. Macro-TSH, although rare, emerged as the key contributor to the patient’s unexplained increase in TSH levels. This case highlights the importance of considering macro-TSH as a potential etiology in cases characterized by unexplained TSH elevation, offering insights into diagnostic protocols and expanding our understanding of thyroid function anomalies.

## 1. Introduction

High thyrotropin (TSH) levels combined with normal free-T3/T4 are usually defined as subclinical hypothyroidism. However, other less common causes, i.e., TSH resistance syndromes, biologically inactive TSH, and laboratory interference, may be considered [1]. More methods for the measurement of TSH have been developed, from immunoradiometric assay (IRMA) to the recent chemiluminescence enzyme immunoassay (CLEIA), chemiluminescence immunoassay (CLIA), and electrochemiluminescence immunoassay (ECLIA). These latter immunoassays provide greater precision, accuracy, and a wider range of measurement [2]. Unfortunately, despite significant methodological improvements, immunoassays used for thyroid tests may still be affected by interferences. Some assay interferences include heterophilic antibodies, anti-mouse antibodies, and macromolecules (e.g., macro-TSH). Macro-molecules are considered biologically inactive, and they attract interests related to circulating hormones. In the field of endocrine disorders, the case of macro-prolactin (macro-PRL) is one of the most frequent. In patients with very high PRL levels in the absence of hyperprolactinemia-related symptoms and pituitary enlargement/lesion, searching for macro-PRL is recommended with the aim of avoiding unnecessary long-term therapy. To detect the presence of macro-PRL, polyethylene glycol (PEG)-mediated precipitation is usually performed, and patients having this condition can be followed up without medications. Since PEG precipitation has a low cost and is reliable for detecting macro-PRL while avoiding unnecessary treatments, it is largely diffused in laboratory departments and largely used in clinical practice. Macro-TSH is a macromolecule produced through the formation of an autoimmune anti-TSH immunoglobulin (Ig) complex with the TSH molecule. Like other macro-molecules, macro-TSH is biologically inactive, and only a small fraction of the total amount of hormones in the plasma remains free [3]. This is why, in the case of macro-TSH, thyroid hormone levels are usually normal. Nonetheless, due to its large molecular size, the clearance of macro-TSH may be delayed, and it can accumulate in blood circulation, leading to elevated serum TSH and a consequent clinical mismanagement. As in the scenario of patients with macro-PRL, finding cases with macro-TSH is useful to avoid unnecessary treatment with levothyroxine (LT4). In fact, macro-TSH presents with a biochemical subclinical hypothyroidism (i.e., high TSH combined with normal free-T3 and free-T4) in the absence of the causes of hypothyroidism, such as positive anti-thyroid auto-antibodies and the alteration of thyroid parenchyma. Typically, and of importance for clinical practice, the TSH of patients with macro-TSH is very high, often above 100 mIU/L, and this presentation probably indicates the need to start LT4 therapy [3,4,5,6,7,8,9]. The gold standard to detect macro-TSH is chromatography [7]. However, using chromatography is quite limited in clinical practice because of its significant costs. Therefore, the literature has debated alternative tests for when chromatography is unavailable. Among these, PEG precipitation has been reported as reliable, but specific cutoffs to discriminate between the presence or absence of macro-TSH are still not available. Since both the clinical significance and prevalence of macro-TSH remain unknown, it was suggested that patients diagnosed with this condition should be followed up through the measurements of TSH, free-T3, and free-T4 to overcome the difficulty of interpreting TSH levels [8].

Here, we report a case investigated at our institution for very high unexplained TSH levels finally diagnosed with macro-TSH using size exclusion chromatography (SEC). The clinical presentation of the patient is fully described. Sequential evaluations performed during clinical practice are reported with their results. A clinically oriented discussion is then addressed to furnish clinicians with useful information when faced with patients suspected of having macro-TSH conditions.

## 2. Case Report

A 69-year-old male patient was referred to the Endocrinology Clinic of EOC in March 2023. Back in 2018, he was treated with an anti-thyroid drug over a 3-month period for amiodarone-induced thyrotoxicosis. Then, amiodarone was discontinued. Later, after a short period of euthyroidism, in 2019 he was treated by a GP with LT4 for newly discovered hypothyroidism, and optimal replacement was recorded up to 2021. Since 2022, the thyroid hormone substitution was suboptimal, requiring more adjustments of the LT4 dose. Combined LT4/LT3 therapy (Novothyral^®^, 2.0 mcg/kg of LT4, 0.4 mcg/kg of LT3) was also used with no significant changes.

During our first visit, the patient presented a history of intermittent atrial fibrillation, obstructive sleep apnea syndrome, and current obesity (BMI 31 kg/m^2^). Non-thyroid therapy included bisoprolol, olmesartanmedoxomil/idroclorotiazide, rivaroxaban, and ezetimibe/simvastatin. Through an ultrasound, the thyroid showed a normal echostructure. Since the TSH value was not extremely high to trigger the immediate conclusion of assay interference (<50 mIU/L), we initially ruled out gastrointestinal malabsorption, and TSHoma was also excluded through magnetic resonance of the pituitary gland. Then, the thyroid therapy was adjusted to LT4/LT3 doses of 1.76 mcg/kg and 0.17 mcg/kg, respectively. TSH was measured using the electrochemiluminescence method on a Cobas^®^ 6000 modular analyzer (Roche Diagnostics International Ltd., Rotkreuz, Switzerland). Since TSH did not change over time (i.e., 36.7 mIU/L, 53.2 mIU/L, 51.7 mIU/L, and 46.3 mIU/L in August 2023), we considered the potential presence of assay interferences and started laboratory work. Firstly, TSH was measured following serum dilutions (1:2 and 1:10) with a value of 48.4 mIU/L (recovery, 104.5%) and 48.3 mIU/L (recovery, 104.3%), respectively. Secondly, the sample was treated to exclude the presence of heterophile antibodies using a Heterophilic Blocking Tube (HBT) (Scantibodies Laboratory, Inc., Santee, CA, USA), resulting in a TSH of 47.5 mIU/L (recovery, 102.6%), thus excluding possible interference. Finally, the sample was treated with PEG precipitation to remove high-molecular-weight proteins, which includes macro-TSH. The result we achieved was a significant reduction in TSH levels to 2.22 mIU/L with a recovery rate of 4.76%. Thus, we concluded that the condition was macro-TSH. Figure 1 illustrates the TSH values recorded during the clinical investigation.

SEC was performed on the patient sample and a control with elevated TSH levels (Figure 2). A plasma sample diluted 1:10 with PBS (pH 7.4) was fractionated on an AdvancedBio SEC 300A coupled with an AdvancedBio SEC 130A column (PBS pH 7.4, 0.2 mL/min) via 1290 HPLC Agilent Technologies. Fractions of 0.5 min were collected, and TSH was analyzed using Cobas 8000 e801 (Roche diagnostics). The TSH of the control patient sample eluted between 48 KDa and 23 KDa according to the theoretical 28 KDa molecular weight, while the case-study patient sample eluted between 223 KDa and 152 KDa according to the sum of the immunoglobulin and TSH (theoretical, 178 KDa).

## 3. Discussion

Macro-TSH is a rare entity and represents a clinical challenge [3,4,5,6,7,8,9]. In particular, its estimated prevalence was reported to be 0.79% [6]. The present case was initially investigated for causes of unexplained elevated TSH as we previously described [10]. After the exclusion of malabsorption and TSHoma, and considering the normal thyroid ultrasound presentation, the possible presence of assay interference was taken into account. The initial assessment of interference was not immediate, which was also due to the absence of an exceptionally high value, which would immediately raise suspicions of laboratory errors. This holds significance because other studies [7,8] have shown higher TSH levels (>100 mIU/L) compared to our case, which could have potentially indicated the interference issue earlier.

Focusing on the laboratory procedures performed, as mentioned before, we first performed a 1:2 and 1:10 serum dilution. The TSH was unchanged, showing that there were no interfering substances in the test, which sometimes happens with drugs. Then, we searched for interference from heterophile antibodies, which can lead to inaccurate results by causing cross-reactivity or interference in immunoassays like the one used to measure TSH. To evaluate the interference caused by heterophile antibodies, we used a heterophile blocking tube, a specialized device typically containing substances that can bind to the heterophilic antibodies and prevent their interaction with the assay components, thus reducing the interference. In our case, the TSH value was again substantially unchanged, thus suggesting excluding heterophile antibodies interference. The blood sample was finally evaluated for the possible presence of macromolecules using the PEG precipitation technique, which is often used to selectively precipitate or separate molecules from a solution based on their size or molecular weight. Indeed, this technique helped to isolate the macro-TSH and associated molecules, leading to a more accurate measurement of the patient’s TSH level. Chromatography finally confirmed the condition of macro-TSH. These techniques used in the lab were crucial to identify macro-TSH, prompting us to revise the therapy. Since hypothyroidism was detected just after the discontinuation of amiodarone, further revisions of the thyroid therapy are planned.

The underlying reason for the interference caused by macro-TSH is rooted in its molecular weight. TSH possesses a molecular weight of around 30 kDa and can be easily filtrated through the kidney. Nevertheless, upon binding with an Ig molecule, TSH forms a larger compound called macro-TSH, with an approximate weight of 200 kDa. This impedes its passage through the kidney’s filtration system, resulting in its buildup within the serum. Even though macro-TSH complex lacks biological activity, it still retains its capacity to provoke an immune response [4]. The concept of macro-TSH is actually derived from the scientific knowledge of macro-PRL, which has drawn even more research attention and occurs with a higher prevalence with respect to macro-TSH, ranging from 9.7% to 29% according to the immunoassay platform employed [11]. The origin of macro-TSH’s appearance continues to be unclear. Similarly, there is insufficient understanding regarding whether macro-TSH’s presence might be prolonged throughout a person’s lifetime or if it is of a transient nature. The research conducted by Hattori et al. established that macro-TSH tends to endure over an extended period [5]. In our specific case, the patient had indeed been affected by macro-TSH for more than one year. It is also probable that, aside from potentially impeding the diagnostic procedure, the existence of macro-TSH likely does not pose any harm to the patient [8]. However, it is worth noting that no routine TSH immunoassay can disclose the presence of macro-TSH [7].

A practical discussion is needed regarding diagnostic procedures of macromolecules. It is worth noting that chromatography stands as the gold standard for diagnosing macro-TSH [7]. Chromatography involves using an acidic elution buffer (pH 3.0) to separate the antigen-antibody complex during gel filtration. However, chromatography is expensive and not routinely available in most institutions [8]. Therefore, also considering the rarity of macromolecules in clinical practice, there is a need for alternative methods when chromatography is unavailable, as in our experience. In the case of macro-PRL, among various techniques that remove it from the serum prior to immunoassay, treatment with PEG is the most commonly employed method [12]. For the diagnosis of macro-PRL, a recovery percentage of 40% has been suggested. Conversely, there is no standardized cutoff value for PEG-precipitated TSH to definitively diagnose macro-TSH, although Loh et al. [8] proposed a diagnosis of macro-TSH when the recovery is less than 20%. Overall, it is important to recognize that chromatography remains the gold standard for diagnosis. Nonetheless, given its limited availability, it is crucial for laboratories to have alternative methods to diagnose macromolecule-related conditions. The herein described case, as well as several other reports, indicates that PEG may serve as a surrogate when chromatography is not accessible, and its results can be strengthened by the consistency of clinical findings, as in the case we described. Confirmation by chromatography still remains mandatory.

## 4. Conclusions

Given the increase in the knowledge of laboratory interferences [13], when a significant rise in TSH is observed and it is unexplained from the clinical standpoint, further investigation should be conducted. It is essential to rule out various sources of interferences. Thyroidologists should be aware that macro-TSH can be present, and no commercial assays can exclude this condition even when TSH is not particularly elevated (i.e., 30 to 50 mIU/L). Laboratory departments should be well prepared to handle cases in whom the possibility of macro-TSH should be ruled out. Recognizing patients with macro-TSH allows them to avoid unnecessary treatment. In this context, chromatography remains the gold standard. A multidisciplinary approach is important to prevent unnecessary and possibly invasive/expensive procedures, as well as misguided and ineffective treatment approaches. Lastly, we prompt clinicians who diagnose these cases to report them. Also, case reports and case series, generally considered as not being high-quality medical literature, can improve the knowledge of the macro-TSH condition and the specific laboratory tests capable of detecting it.

## Figures and Tables

**Figure 1 jpm-13-01471-f001:**
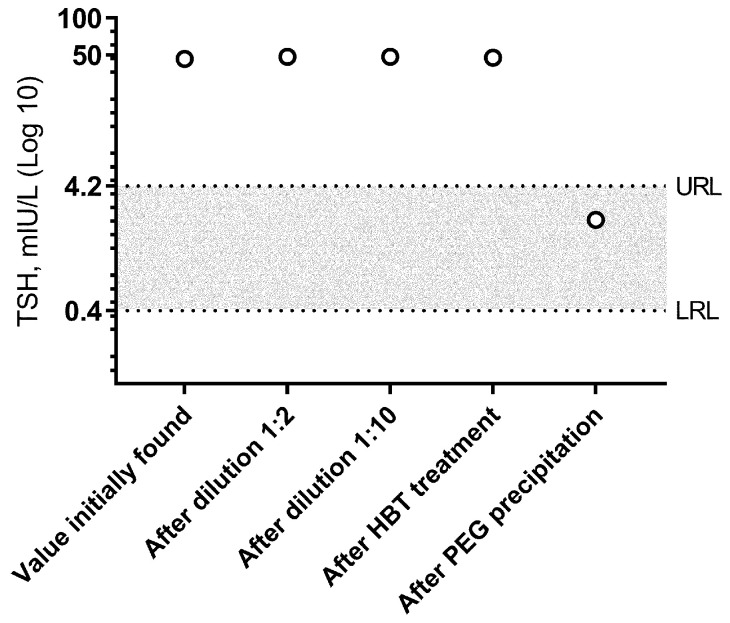
Values of TSH observed in the present case. Levels of TSH (**○**) after dilution represent the values obtained after multiplication with the dilution factor. HBT, heterophil blocking tube. PEG, polyethylene glycol. URL, upper reference limit. LRL, lower reference limit.

**Figure 2 jpm-13-01471-f002:**
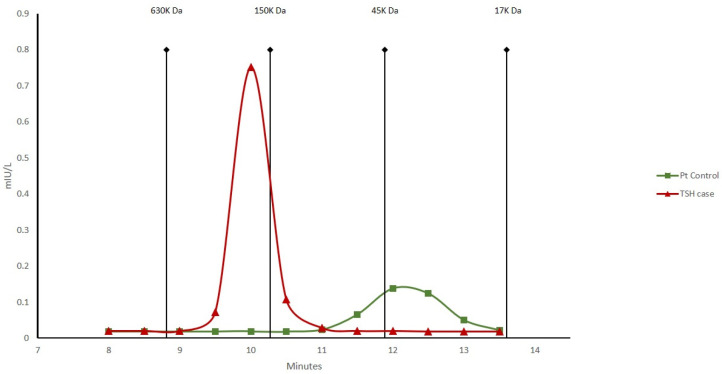
SEC analysis of TSH of control patient sample and case-study patient. Total of 20 µL of plasma diluted 1:10 with PBS pH7.4. Fraction of 0.5 min was analyzed with immunoassay for TSH. The chromatogram was calibrated with the AdvancedBio SEC 300A protein standard (Agilent Technologies). Molecular mass of calibrants is indicated.

## Data Availability

Data sharing is not applicable.

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
