# Peer review of "Macro-TSH: An Uncommon Explanation for Persistent TSH Elevation That Thyroidologists Have to Keep in Mind"

_jpm, 2023, doi:10.3390/jpm13101471_

Round 1

Reviewer 1 Report

Chiardi et al. have nicely presented a case report of macro-TSH causing elevated TSH in a euthyroid patient. The case is of interest given the rarety of the condition and especially considering the only modestly elevated TSH measured.

The authors conducted 2-fold and 10-fold dilution, used heterophile blocking tube and PEG-precipitation to arrive at the conclusion of the presence of macro-TSH. While PEG-precipitation may be used to strengthen the suspicion of macro-TSH, it is recommended that macro-TSH is confirmed using gel filtration chromatography which is considered the gold standard [Larsen et al. Eur Thyroid J 2021;10:93–97].  

For a case study on macro-TSH to be published, this reviewer would request the conclusion of macro-TSH be confirmed using the gold standard of gel filtration chromatography.

Author Response

We really thank the Reviewer who appreciates how the case was presented. We fully agree with the Reviewer comment about the issue that chromatography represents the god standard to diagnose macro-TSH. However, as largely known, this technique is expensive and often unavailable, as in our institution. Also, cases with suspicious macro-TSH are infrequent. Then, alternatives are warrented. To highlight these issues, a clinical practice-based discussion has been added (in red color in the text), the role of chromatography as gold standard has been underlined, and the usefulness of PEG as alternative option has been discussed. According to this additional discussion, conclusion have been extended. We feel that these issues can be useful for clinicians facing this kind of patient during their clinical practice.

Reviewer 2 Report

The manuscript submitted by Dr. Chiardi et al. is a concise case report describing a real example of personalized medicine, i.e. situation when the lab results did not match patient's clinical status. In their paper authors present a 69-year-old male with persistently elevated TSH levels, ranging from 40 to 50 mU/L , despite lack of clinical symptoms of thyroid disese. The diagnostic challenge was even greater because of previous history of thyroid disorders, fluctuating from hyper to hypothyroidism due to amiodarone. In the presented case the authors considered all options and described them step-by-step in a clear way. I believe that this short case report may be of interest for clinicians who provide care to patients with altered thyroid function tests.

Author Response

We really thank the Reviewer who appreciated the paper.

Round 2

Reviewer 1 Report

The authors have revised several sections of the manuscript for a new focus on the two methods for diagnosing macro-TSH: Gel filtration chromotography and PEG precipitation.

The authors state that the gold standard, gel filtration chromotography, is exprensive and unavailable at the host instition and given the infrequency of macro-TSH, PEG precipitation. must be accepted as an alternative method.

This reviewer agrees that more information is needed on the cutoffs when using PEG precipitation as aluted to by the authors in the manuscript. But contrary to the authors, I find this is a further arguement to perform gel filtration chromotography for confirmation of the performed PEG precipitation. While PEG precipitation can be acceptable in daily clinic, when one wished to publish a case on a rare diagnostic challenge, one must also use the gold standard for confirmation. Therefore, I strongly encourage the authors to seek scientific collaboration to perform the confirmatory gel filtration chromotography.   

Author Response

We totally agree with the Reviewer that the confirmatin by chromatography is needed when a case is published. Accordingly, our laboratory department made effort and performed that. Chromatography confirmed the condition of macro-TSH. The text has been revised and changes appear in red. The colleague who performed chromatography has been added as co-author. We really thank the Reviewer who prompted us to better define the diagnosis.

Round 3

Reviewer 1 Report

The authors have conducted the confirmatory analysis as suggested which greatly increases the quality of the report.